# Microwave-Assisted Synthesis of Mono- and Disubstituted 4-Hydroxyacetophenone Derivatives via Mannich Reaction: Synthesis, XRD and HS-Analysis

**DOI:** 10.3390/molecules24030590

**Published:** 2019-02-07

**Authors:** Ghadah Aljohani, Musa A. Said, Dieter Lentz, Norazah Basar, Arwa Albar, Shaya Y. Alraqa, Adeeb Al-Sheikh Ali

**Affiliations:** 1Chemistry Department, College of Science, Taibah University, P.O. Box 30002, Al-Madinah Al-Munawarah 14177, Saudi Arabia; musa_said04@yahoo.co.uk (G.A.); shaya97@hotmail.com (S.Y.A.); 2Institut für Chemie und Biochemie, Anorganische Chemie, Freie Universität Berlin, Fabeckstr. 34-36, 14195 Berlin, Germany; dlentz@zedat.fu-berlin.de; 3Department of Chemistry, Faculty of Science, Universiti Teknologi Malaysia, Johor Bahru 81310, Malaysia; norazahb@utm.my; 4Department of Physics, College of Science, Jeddah University, P.O. BOX 80327, Jeddah 21589, Saudi Arabia; arwa.albar@kaust.edu.sa

**Keywords:** Mannich bases, 4-hydroxyacetophenone, microwave irradiation, regioselectivity, X-ray, HSA

## Abstract

An efficient microwave-assisted one-step synthetic route toward Mannich bases is developed from 4-hydroxyacetophenone and different secondary amines in quantitative yields, via a regioselective substitution reaction. The reaction takes a short time and is non-catalyzed and reproducible on a gram scale. The environmentally benign methodology provides a novel alternative, to the conventional methodologies, for the synthesis of mono- and disubstituted Mannich bases of 4-hydroxyacetophenone. All compounds were well-characterized by FT-IR, ^1^H NMR, ^13^C NMR, and mass spectrometry. The structures of 1-{4-hydroxy-3-[(morpholin-4-yl)methyl]phenyl}ethan-1-one (**2a**) and 1-{4-hydroxy-3-[(pyrrolidin-1-yl)methyl]phenyl}ethan-1-one (**3a**) were determined by single crystal X-ray crystallography. Compound **2a** and **3a** crystallize in monoclinic, *P*2_1_/*n*, and orthorhombic, *Pbca*, respectively. The most characteristic features of the molecular structure of **2a** is that the morpholine fragment adopts a chair conformation with strong intramolecular hydrogen bonding. Compound **3a** exhibits intermolecular hydrogen bonding, too. Furthermore, the computed Hirshfeld surface analysis confirms H-bonds and π–π stack interactions obtained by XRD packing analyses.

## 1. Introduction

Mannich bases can be easily transformed into numerous compounds due to their high reactivity. Several Mannich bases are of particular interest due to their importance as precursors of valuable pharmaceuticals products and their use as synthetic building blocks [1,2]. Mannich bases have attracted more and more attention from chemists due to their specific biological activities, such as antiviral [3,4], acetylcholinesterase inhibitory [5,6,7], antioxidant [8,9], and antiproliferative [10,11] activity. Mannich bases are also key precursors to accessing amino alcohols, peptides, lactams, and optically active amino acids [12,13]. Consequently, many researchers were attracted to investigating green protocols for the Mannich reaction during the last period [2]. The Mannich reaction is an essential reaction for the direct formation of new C–C and C–N bonds [10,14]. The products of Mannich reaction of 4-hydroxyacetophenone as substrate vary depending on the reaction conditions. Using classical conditions, such as low pH, the acetyl group undergoes reaction, producing β-amino alkyl ketone. In this case, the substrate is converted into the corresponding Mannich base through an aminomethylation process. On the other hand, under an alkaline medium, the Mannich reaction is selectively undergoes electrophilic aromatic substitution in the *ortho* position to the hydroxyl group (Scheme 1) [1]. The conventional multicomponent Mannich reactions are found to have drawbacks such as a long reaction time, harsh reaction conditions, and toxic vapor and byproducts [15]. As a consequence, different reaction conditions have been widely described for Mannich reaction in the presence of a variety of catalysts [16,17]. Additionally, a continuous search for the synthetic methodologies of cleaner Mannich reactions seeking green chemistry principles includes the use of protocols in water [14], biodegradable and reusable catalysts [18,19], ionic liquids [12,19,20], and solvent-free and/or catalyst-free media reaction conditions under microwave (MW) and ultrasound irradiation [21,22,23,24,25,26,27]. Microwave irradiation has been used to synthesize a lot of organic compounds. In fact, the use of microwave irradiation offers a higher reaction rate and better product in chemical synthesis [28], however, few studies have been done utilizing microwave irradiation in the synthesis of Mannich bases [21,25]. These studies were done under acidic conditions using substituted acetophenones whereas the regioselectivity of the reaction was studied on the 2-hydroxy-chalcone backbone [29]. Hence, we describe, herein, the first efficient, gram-scale, non-catalyzed and microwave-assisted synthetic approach of mono- and disubstituted Mannich bases of *para*-hydroxyacetophenone with various secondary amines.

## 2. Results and Discussion

### 2.1. General Synthesis

Firstly, the reactions were run in small scales (2 mmol) of 4-hydroxyacetophenone with different ratios of morpholine as a model reaction, to find out the optimal conditions in terms of solvent, time, reactant ratio, and temperature. Four different solvents, namely, water, ethanol, ethylene glycol, and 1,4-dioxane, at fixed power (300 W) were used for optimization. The temperature ranged between 100 and 120 °C, higher than the boiling points of the chosen solvent as the experiments were done in a sealed vessel. When water was used as a solvent, the corresponding product was not observed based on the TLC monitoring and the reaction led to the formation of an insoluble product in classical organic solvents. On the other hand, using polar protic solvents (ethanol or ethylene glycol) gave the corresponding monosubstituted Mannich product **2a** at a very low yield. Interestingly, an excellent yield was obtained when 1,4-dioxane was used as a solvent. The TLC showed complete consumption of 4-hydroxyacetophenone after 30 min by disappearance of the starting material spot, and two new main spots were observed using 1.5 equivalents of formaldehyde and morpholine as a model for optimization (Scheme 2).

The resulting mixture was purified via column chromatography. The first eluted compound was characterized using spectroscopic analysis. The ^1^H NMR analysis of compound **2a** showed the disappearance of one of the aromatic protons, revealing regioselective monosubstitution on the aromatic ring. ^1^H NMR exhibits a broad singlet at 3.78 ppm integrated to six protons related to the benzylic protons which overlapped with four protons in (OCH_2_) related to morpholine moiety. A multiplet peak, between 2.54 and 2.59 ppm, was assigned to the protons in (NCH_2_) in the morpholine moiety and the methyl group. The ^13^C spectrum verified the formation of Mannich base **2a** due to the presence of three peaks of CH_2_ group related to (NCH_2_) at 52.74 ppm, benzylic (CH_2_) at 61.24 ppm, and (OCH_2_) of morpholine moiety at 66.52 ppm. Moreover, the presence of only three resonances for the aromatic CH at 116.07, 129.75, and 130.56 ppm confirmed the formation of monosubstituted Mannich base. On the other hand, the ^1^H NMR spectral analysis of the second compound showed the formation of disubstituted Mannich product **2b**. As a result, two protons disappeared in the aromatic region and the integration of two morpholine moieties to the other protons confirmed the formation of **2b**. The ^13^C exhibit a single resonance of two equivalent aromatic CH at 130.24m confirming the occurrence of disubstitution reaction at the benzene ring. Meanwhile, the mass spectral data are in agreement with the molecular formula of **2a** and **2b**. Due to the successful results, a gram-scale (16 mmol) experiment was run for morpholine to determine the overall yield of both **2a** and **2b**.

Comparing the conventional method to the microwave radiation, it was found that the reaction time decreased considerably from 22 h, using thermal heating, to 30 min in order to furnish mono- and/or disubstituted Mannich bases **2a** and **2b** in quantitative yield [30]. Using the same optimized conditions (power, solvent, temperature, and reactant ratio) for the formation of mono- or disubstituted Mannich bases, compounds **3a**–**5a** and **3b**–**4b** were synthesized after 15 min in overall yields of 64% to 77% (Table 1, Appendix A). The time was monitored for each amine based on TLC results.

In order to improve the product ratios, different reactant ratios were used in 1,4-dioxane at 120 °C (which was found to be the optimal reaction temperature) and 300 W. It was found that the monosubstituted Mannich base **2a** was obtained as a major product (80% after column) when 1.5 equivalent of the secondary amine and formaldehyde were used. However, the disubstituted Mannich base **2b** was formed quantitively as the only product when 2 equivalents of the secondary amine and formaldehyde were used (Table 1, Appendix A). The product yield was greatly increased in quantitative yield, compared to the 60% that had been obtained before [30]. For the novel compounds **3b** and **4b**, the yield under the same conditions was 86% and 94%, respectively. It should be noted that the diethylamine used did not undergo disubstituted Mannich reaction under the same conditions and only monosubstituted **5a** was obtained but in nearly quantitative yield. This may be due to the free rotation of diethyl groups that might sterically prevent the occurrence of disubstituted reaction in comparison to the rigid cyclic secondary amines.

### 2.2. Crystal Structure Description

The structures of the monosubstituted compounds **2a** and **3a** were proposed on the basis of ^1^H and ^13^C NMR, and IR spectroscopic analyses. In order to gain information on their structural characteristics, X-ray crystallographic analyses were carried out for both **2a** and **3a** [31,32,33,34,35,36,37]. The molecular structures and some atom-numbering schemes are depicted in Figure 1 and Figure 2 for **2a** and **3a** respectively. Crystallographic data and structure refinement parameters for **2a** and **3a** are listed in Table 2. Selected bond lengths, angles, and torsion angles are listed in Table 3 and Table 4. The most characteristic feature of the molecular structure of **2a** is that the morpholine fragment adopts a chair conformation with strong intramolecular hydrogen bonding. One carbon atom of the chair conformation deviates from the main plane by only 0.015 Å, indicating an almost perfect chair conformation for the morpholine fragment. The acetyl substituent is located almost in the plane of the aromatic with a torsion C5–C4–C12–O3 of 1.6°. However, the morpholino substituent is turned out of the ring plane resulting in C1–C2–C7–N1 of −45.7°. There exists only one intramolecular hydrogen bond of H1 to the nitrogen atom N1 (O1…N1 2.634 (2) with a normalized [38,39] O–H of 0.938 Å, resulting in a N…H distance of 1.801 and an O1–H1–N1 angle of 146.5°). No intermolecular hydrogen bonds can be detected for **2a**.

By contrast, compound **3a** exhibits intermolecular hydrogen bonding. Another interesting structural aspect in **3a** is related to the molecular packing stability. It is documented that intermolecular hydrogen bonding plays a significant role in the assembling of molecules into a well-ordered crystal packing structure. This is because intermolecular hydrogen bonding helps to form a network (Figure 3) [40,41]. It is worth mentioning that heterocycles containing a pyrrolidine moiety are frequently found in many bioactive compounds [42]. The pyrrolidine ring in **3a** has a slightly twisted envelop conformation with a deviation of the C11 atom from the main plane of C8, C9 and C10 by 0.234 Å (Figure 2), indicating that the pyrrolidine ring has some ring strain [42]. The N1 atom is found above this main plane by 0.444 Å (Figure 2). On the other side, the angles around N1 of 114.80°, 103.19°, and 112.90° indicating a distorted tetrahedral geometry around the central atom N1.

### 2.3. Hirshfeld Analysis

The Hirshfeld surface analysis (HSA), mapped over d_norm_ and shape index plots, are illustrated in Figure 4a,b for compounds **2a** and **3a**, respectively. Table 5 summarizes the relative contributions of different atomic contacts to the HSs. The d_norm_ surface of **2a** compound (ranging from –0.259 Å to 1.364 Å) shows two deep red spots related to H…O (2.267 Å) and H…H (2.767 Å), indicating strong short contacts. The H…H interaction is found to be dominating for the **2a** compound, accounting for 55.8% of HS area, whereas H…O/O…H contact accounted for 12.6% and 14.8% of HS area. The white spots on HS are mainly due to H…H contact. Shape index plot for **2a** compound shows adjacent red and blue triangle (red-dash circle) indicating the presence of π–π stacking. The d_norm_ surface of **3a** (ranging from –0.758 Å to 1.239 Å) shows two deep red spots related to O–H…N/N…H–O strong short contacts (due to distances of 1.657 Å/2.132 Å), which accounted for only 1.6% of HS area. Most of the white spots on the shape index surface are due to H…H contacts, which contribute with 60.7% to HS area. The zero contribution of C…C interaction to HS for **3a** compound indicates the absence of π–π stacking. This is in agreement with that shown by XRD. The π–π stacking distance, in **3a**, is measured as the distance between the aromatic centroids. The X-ray structure of **3a** shows a separation of 6.183 Å between the aromatic centroids. Shape index plots for **2a** and **3a** show red concave and blue convex areas, which characterize the packing mode with nearby molecules (Figure 4).

Figure 5 shows the 2D fingerprint plots for the overall compounds, as well as for specific contacts. The H…H interaction is indeed the dominating one for **2a** and **3a** compounds. The comparable wings of **2a** and **3a** fingerprint plots, which are due to C…H/H…C contacts, indicate similarity between the two compounds. C…H/H…C contact acts as a secondary interaction with no significant peaks. For **2a**, the red-dash circle indicates the 0.9% contribution of π–π stacking. While O–H…N/N…H–O contact plays a critical role in **3a** compound, it is completely absent in the **2a** compound. The two small side peaks obtained for **2a** compound are due to O…H/H…O contact with tips at d_i_ + d_e_ =2.25 Å. However, the shortest contact is obtained by the middle peaks (with tips at d_i_ + d_e_ =2.2 Å), which are due to H…H contact, indicating strong H…H interaction. Interestingly, two distinct sharp spikes are obtained for **3a**, which are due to considerably short O–H…N/N…H–O contacts, indicating strong molecular interaction. The corresponding d_i_ and d_e_ combination are 0.65 Å and 1.05 Å with tips at d_i_ + d_e_ =1.7 Å. Consequently, HAS findings are in agreement with XRD packing analyses.

## 3. Materials and Methods

All solvents and reagents were purchased from Sigma-Aldrich (Deisenhofen, Germany) and used without further purification. 4-Hydroxyacetophenone was purchased from Acros (Thermo Fisher Scientific Geel–Belgium). The monitoring of reaction was done by the utilization of pre-coated silica gel plates (60 F_254_) and thin layer chromatography (TLC). The normal phase silica gel (Merck, 70–230 mesh) was used to perform column chromatography purification, while the Merck silica gel (230–400 mesh) was utilized to perform the vacuum liquid chromatography.

A controllable single-mode microwave reactor, CEM Discover Microwave™ designed for synthetic utilization, was used. The reactor consists of power and pressure controls in addition to a magnetic stirrer. The temperature is controlled by an IR sensor. Melting points were measured using Sanyo MPD350 apparatus (Gallenkamp, Osaka, Japan) with digital display, and they were not corrected. A Perkin Elmer ATR spectrophotometer (Waltham, MA, USA) was used to record the infrared (IR) spectra. A Bruker Advance 400 MHz spectrometer (Fällanden, Switzerland) was used to record ^1^H NMR and ^13^C NMR spectra. NMR samples were separately measured in DMSO, CDCl_3_, and MeOD at room temperature. Mass spectra were recorded on Finnigan MAT95XL (ThermoFisher Scientific, Bremen, Germany) using (EI), at 70 eV.

### 3.1. General Synthesis of Mannich Bases (***2***–***5***)

To a solution of 4-hydroxy-acetophenone (**1**) (16 mmol, 2.18 g) and formaldehyde (24 mmol or 32 mmol) in 1,4-dioxane (15 mL), the corresponding secondary amine (morpholine (**2**), pyrrolidine (**3**), piperidine (**4**) or diethylamine (**5**)) was added using the same ratio of formaldehyde. This mixture was placed in the MW sealed vessel with stirring. The reaction mixture was irradiated for 15–30 min at 120 °C (power 300 W). TLC was used to monitor the progress of the reaction. After consumption of the starting materials, the vessel was removed and cooled down to room temperature. The reaction mixture was concentrated under reduced pressure and purified using column chromatography.

#### 3.1.1. 1-{4-Hydroxy-3-[(morpholin-4-yl)methyl]phenyl}ethan-1-one (**2a**)

The purification of the crude product, which has been synthesized in the respective ratio of 1:1.5:1.5 of compound **1**, formaldehyde, and morpholine, was done using column chromatography with hexanes/EtOAc (8:2) as eluent to yield **2a** as colorless crystals (3.02 g; 80%), mp 69–70 °C. IR spectrum (ν_max_/cm^−1^): 1110 (C–N), 1246 (C–O), 1581 (C=C), 1666 (C=O), 2978 (CH_3_, sp^3^). ^1^H NMR (ppm/δ, 400 MHz, CDCl_3_): 2.59–2.54 (m, 7H), 3.77 (br s, 6H), 6.85 (d, *J* = 8.5 Hz, 1H), 7.69 (s, 1H), 7.82 (d, *J* = 8.5 Hz, 1H), 11.32 (br s, 1H, –OH). ^13^C NMR (ppm/δ, 100 MHz, CDCl_3_) δ 26.24, 52.74, 61.24, 66.52, 116.07, 120.23, 129.20, 129.75, 130.56, 162.43, 196.77. EIMS, *m*/*z* (% rel. intensity): 235(100) [M^+^, C_13_H_17_NO_3_], 204 (16), 188 (38), 176 (12), 162 (20), 149 (76), 133 (18), 106 (10), 100 (10), 86 (45), 77 (21), 57 (22).

#### 3.1.2. 1-{4-Hydroxy-3,5-bis[(morpholin-4-yl) methyl]phenyl}ethan-1-one (**2b**)

The purification of the crude product which has been synthesized in the respective ratio of 1:2:2 of compound **1**, formaldehyde, and morpholine was done using column chromatography with EtOAc/MeOH (9:1) as eluent to yield **2b** as colorless needles (5.32 g; quantitative yield) mp 88–90 °C after removing the solvent. IR spectrum (ν_max_/cm^−1^): 1113 (C–N), 1304 (C–O), 1597(C=C), 1669(C=O), 2958 (CH_3_, sp^3^). ^1^H NMR (ppm/δ, 400 MHz, MeOD): 2.59–2.55 (m, 11H), 3.74–3.72 (m, 12H), 7.83 (s, 2H), 7.92 (s, 1H). ^13^C NMR (ppm/δ, 100 MHz, MeOD): 24.90, 52.78, 58.18, 66.32, 121.97, 128.25, 130.24, 161.44, 197.94. EIMS, *m*/*z* (% rel. intensity): 334(14) [M^+^, C_18_H_26_N_2_O_4_], 276 (6), 247 (100), 217 (12), 189 (14), 162 (8), 133 (34), 119 (11), 86 (24), 56 (14).

#### 3.1.3. 1-{4-Hydroxy-3-[(pyrrolidin-1-yl) methyl]phenyl}ethan-1-one (**3a**)

A ratio of 1:1.5:1.5 of compound **1**, formaldehyde, and pyrrolidine, respectively, were used to synthesize **3a**. Eluting system of hexanes/EtOAc (6:4), was used to purify the crude product to yield pale-yellow crystals, (1.72 g; 49%), mp 92–95 °C. IR spectrum (ν_max_/cm^−1^): 1191 (C–N), 1284 (C–O), 1595 (C=C), 1661 (C=O), 2965 (CH_3_, sp^3^). ^1^H NMR (ppm/δ, 400 MHz, MeOD): 1.99–1.94 (m, 4H), 2.51 (s, 3H), 2.92 (t, *J* = 7.0 Hz, 4H), 4.03 (s, 2H), 6.72 (d, *J* = 8.5 Hz, 1H), 7.79 (d, *J* = 2.3 Hz, 1H), 7.83 (dd, *J* = 8.7 and 2.4 Hz, 1H). ^13^C NMR (ppm/δ, 100 MHz, MeOD): 23.02, 24.67, 52.88, 57.13, 116.53, 121.21, 126.14, 130.22, 130.73, 167.15, 197.75. EIMS, *m*/*z* (% rel. intensity): 219 (13) [M^+^, C_13_H_17_NO_2_], 149 (5), 133 (3), 106 (2), 91 (2), 84 (9), 77 (4), 70(100), 51 (2).

#### 3.1.4. 1-{4-Hydroxy-3,5-bis[(pyrrolidin-1-yl) methyl] phenyl} ethan-1-one (**3b**)

The purification of the crude product which has been synthesized in the respective ratio of 1:2:2 of compound **1**, formaldehyde and pyrrolidine was done using column chromatography with EtOAc/MeOH (9:1) as eluent to yield **3b** as dark red oily liquid (4.2 g; 86%). IR spectrum (ν_max_/cm^−1^): 1302 (C–O), 1356 (C–N), 1597 (C=C), 1665 (C=O), 2963 (CH_3_, sp^3^). ^1^H NMR (ppm/δ, 400 MHz, DMSO-*d*_6_): 1.76 (br s, 8H), 2.47 (s, 3H), 2.57 (br s, 8H), 3.77 (s, 4H), 7.69 (s, 2H). ^13^C NMR (100 MHz, DMSO) ppm: δ 23.66, 26.67, 53.54, 55.59, 123.68, 127.57, 128.98, 162.09, 196.56. EIMS, *m*/*z* (% rel. intensity): 302 (14) [M^+^, C_18_H_26_N_2_O_2_], 231 (100), 216 (9), 188 (12), 159 (14), 147 (7), 133 (20), 119 (11), 84 (11), 70 (43), 55 (4).

#### 3.1.5. 1-{4-Hydroxy-3-[(piperidin-1-yl) methyl] phenyl} ethan-1-one (**4a**)

The crude product of **4a** was obtained from the reaction of the respective ratio 1:1.5:1.5 of compound **1**, formaldehyde, and piperidine. The obtained product was purified by using hexanes/EtOAc (6:4) as an eluent to yield **4a** as colorless crystals (2.15 g, 58%), mp 82–83 °C. IR spectrum (ν_max_/cm^−1^): 1256 (C–N), 1287 (C–O), 1594 (C=C), 1661 (C=O), 2946 (CH3, sp^3^). ^1^H NMR (ppm/δ, 400 MHz, CDCl_3_): 1.67–1.52 (m, 6H), 2.53 (br s, 7H), 3.75 (s, 2H), 6.84 (d, *J* = 8.4 Hz, 1H), 7.68 (s, 1H), 7.80 (d, *J* = 8.4 Hz, 1H), 11.41 (br s, 1H). ^13^C NMR (ppm/δ, 100 MHz, CDCl_3_): 23.71, 25.6, 26.22, 53.73, 61.51, 115.93, 121.06, 128.73, 129.35, 130.24, 163.32, 196.86. EIMS, *m*/*z* (% rel. intensity): 233 (10) [M^+^, C_14_H_19_NO_2_], 149 (4), 133 (2), 106 (1), 98 (5), 84 (100), 77 (3), 56 (2).

#### 3.1.6. 1-{4-Hydroxy-3,5-bis[(piperidin-1-yl) methyl] phenyl} ethan-1-one (**4b**)

The crude product resulted from the reaction of the respective ratio 1:2:2 of compound **1**, formaldehyde, and piperidine was purified using hexanes/EtOAc (3:7) as eluent to yield **4b** as pale-yellow crystals mp 93–96 °C (5 g, 94%). IR spectrum (ν_max_/cm^−1^): 1300 (C–O), 1349 (C–N), 1594 (C=C), 1669 (C=O), 2925 (CH_3_, sp^3^). ^1^H NMR (ppm/δ, 400 MHz, DMSO): 1.53–1.42(m, 12H), 2.46–2.43 (m, 11H), 3.6 (s, 4H), 5.51 (br s, 1H), 7.68 (s, 2H). ^13^C NMR (ppm/δ, 100 MHz, DMSO): 24. 13, 25.90, 26.65, 53.84, 58.65, 123.01, 127.92, 129.27, 161.97, 196.59. EIMS, *m*/*z* (% rel. intensity): 330 (14) [M^+^, C_20_H_30_N_2_O_2_], 245 (100), 228 (4), 202 (10), 147 (4), 133 (13), 119 (4), 98 (4), 84 (24), 55 (3).

#### 3.1.7. 1-{3-[(Diethylamino) methyl]-4-hydroxyphenyl} ethan-1-one (**5a**)

The crude product of **5a** was obtained from the reaction of the respective ratio 1:2:2. of compound **1**, formaldehyde, and diethylamine. The product was purified by column chromatography using hexanes/EtOAc (6:4) to yield **5a** as a yellow oily liquid (3.53 g, quantitative yield). IR spectrum (ν_max_/cm^−1^): 1280 (C–N), 1357 (C–O), 1589(C=C), 1667(C=O), 2972 (CH_3_, sp^3^). ^1^H NMR (ppm/δ, 400 MHz, CDCl_3_): 1.07 (t, *J* = 7.0 Hz, 6H), 2.46 (s, 3H), 2.59 (q, *J* = 7.0 Hz, 4H), 3.77 (s, 2H), 6.74 (d, *J* = 8.4 Hz, 1H), 7.60 (s,1H), 7.73 (d, *J* = 8.4 Hz, 1H), 11.47 (br s, 1H). ^13^C NMR (ppm/δ, 100 MHz, CDCl_3_): 10.99, 26.08, 46.22, 56.61, 115.87, 121.68, 128.51, 128.95, 130.06, 163.77, 196.74. EIMS, *m*/*z* (% rel. intensity): 221 (5) [M^+^, C_13_H_19_NO_2_], 206 (6), 149 (8), 132 (2), 106 (1), 77 (2), 58 (100).

### 3.2. X-ray Structure Determination

Crystal data, data collection, and structure refinement details are summarized in Table 2. Crystallographic data for **2a** and **3a** were collected on Bruker *D8 Venture* diffractometer; cell refinement—Bruker *SAINT*, data reduction—Bruker *SAINT* [31]; program(s) used to solve and refine structures—SHELXT 2014/5 and SHELXL2014/7 [32,33]; molecular graphics—ORTEP [34,35]; and software used to prepare material for publication—WinGX [37]. All non-hydrogen atoms were refined anisotropically. The hydrogen atoms at the oxygen atoms were located from Fourier map, at carbon and nitrogen atoms placed in calculated positions and refined as rigid atoms. All calculations were performed on PC using WinGX program. Data collection: images were indexed, integrated, and scaled using the APEX3 data reduction package [31]. All figures were made using the program ORTEP and Diamond program [32,33,34,35,36].

### 3.3. Computations

Hirshfeld surface analysis (HSA) and 2D fingerprint plots calculations were carried via d_norm_ surface property with rescale using CRYSTAL EXPLORER 3.0 program [43]. Normalized distances, d_norm_, based on d_i_ and d_e_, are calculated for each point on the surface. HSA mapped over d_norm_ generates a surface with red, blue, and white areas—red spots correspond to closer contacts (the sum of van der Waals radii and d_norm_ is negative), blue spots correspond to longer contacts (the sum of van der Waals radii and d_norm_ is positive) and white spots correspond to d_norm_ values around van der Waals separations. The 2D fingerprint plots provide information about the frequency of each d_i_ and d_e_ combination across all points on HS. The green, blue, and gray colors correspond to interactions with large contribution, with moderate contribution, and without contribution to HS area, respectively. The identical features along 2D plots diagonal correspond to mirrored internal–external atomic contacts (e.g., O…H and H…O contacts).

## 4. Conclusions

It can be concluded that this novel alternative methodology offers significantly less synthesis time with competitive product yield of Mannich bases in the absence of any catalyst. The X-ray crystal structure of **2a** shows an intramolecular hydrogen bonding and displays a chair conformation for the morpholine fragment and, unlike **3a**, exhibits intermolecular hydrogen bonding. The HAS shows that the intermolecular interactions are dominated by H…H interaction for both compounds, though for **2a** it is stronger. Moreover, 2D fingerprint plots confirm that π–π stacking is only present for **2a**.

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
