# Peer review of "Microwave-Assisted Synthesis of Mono- and Disubstituted 4-Hydroxyacetophenone Derivatives via Mannich Reaction: Synthesis, XRD and HS-Analysis"

_molecules, 2019, doi:10.3390/molecules24030590_

Round 1

Reviewer 1 Report

The paper by Ali et al deals with the microwave synthesis of mannich bases followed by single crystals X ray analysis of two compounds.

First of all, I would like to says that using microwave irradiation is not always green! In this particular case, the authors reduced the reaction time from 22h to 30 minutes (which is a great improvement) but they used different stoichiometry (1.5 eq for amine and formaldehyde vs only 1) compared to cited previous work (ref 38). Excess reagent is definitively not green. The title should be modified.

In the title, authors speak about regioselectivity which is known for this kind of compound and classical. Moreover, regioselectivity is not discussed in the manuscript. The title should be modified.

At the end of the introduction, authors argue that the paper describe large scale reaction. In the synthesis section (P13), they present the synthesis of 2a which yield to 3g. In the paper by Wu et al (ref 38) they obtained 19g of the same compound. This is not large scale

P3 : general synthesis : authors says they run reactions in small scales which is highly imprecise (like large scale). Authors should precise the scale in grams (mg?) or mole. Moreover, authors have to mention that reactions were performed in sealed vessel to hemp non-specialist reader. were the small scales experiment performed with all the amine or only with morpholine. What is the fixed time used for optimization? Those points need to be clarified. Moreover, when dealing with optimization a table with all experiments is greatly appreciated.  

P3 : When using water authors obtained an “insoluble product”. Insoluble is not precise. Insoluble in water? In DCM? In DMSO? Authors should add “insoluble in classical organic solvents”.

Excellent yield was obtained with dioxane, which is not a green solvent… (see above).

Page 4 : authors used 1H and 13C NMR to characterize compound. Spectra of all compounds must be provided in supplementary data to evaluate purity and to follow authors explanations on disappearance of signals.

P5 : “Comparing the conventional method to the microwave radiation, it was found that the reaction time decreased considerably from 22 h using thermal heating to 30 min in order to furnish  mono and or disubstituted Mannich bases 2a and 2b in overall yield 99.9%” : I don’t really understand the sentence. Is the overall yield for mono or disubstituted reaction? (“mono and or disubstituted” is not clear). How is calculated 99.9 %? A table with conditions and yield for “compound a” and “compound b” could help to understand “15 minutes overall yield 64, 72, 77%”.

P5 : “In order to improve the products ratio, different reactant ratios were used in 1,4-dioxane at 120°C 98 (which was found to be the optimal reaction temperature) and 300 W.” When dealing with optimization a table is needed with time, temperature, solvent, stoichiometry, yield etc.

 P5 : “For the novel compounds 3b and 4b the yield under the same 104 conditions was 85.7% and 94.3%, respectively. It should be noticed that using diethylamine did not 105 undergo disubstituted Mannich reaction under the same conditions but only mono substituted 5a 106 was obtained in an excellent yield 99.7%” : Table is needed.

Diethylamine do not provide disubstituted compound. Authors should provide a tentative of explanation.

P12 : materials and methods : mass spectrometer is not described (model).

Authors describes acronyms (TLC, CC, VLC) and didn’t use them. Thinlyaer in two words.

P13 : “without KBr” is not useful since authors used ATR technic.

P13 : general synthesis of mannich bases : authors should precise sealed vessel instead of capped with rubber cap since high pressure. At first sight all the reactions were run on 16 mmol scale (ie 2.18 g). quantity of reactant should be included in experimental section and not only equivalents. Yields should be uniformized in all the manuscript. Sometime there is no digits after the decimal point, sometime one, sometime 2. Not sure that 2 digits is representative but authors have to choose between zero and one digit. Same remark with mass of product.

P13 : 1-{4-hydroxy-3-[(morpholin-4-yl) methyl] phenyl} ethan-1-one (2a) : 1H NMR description is different from ref 38. Spectra must be provided.

P14 : “1-{4-hydroxy-3,5-bis[(morpholin-4-yl) methyl] phenyl} ethan-1-one (2b)” : 2b instead of 2a

There is a problem with the yield. 5.39 g correspond to 16.1 mmol (MW of 2b = 334.4) which is superior to the 16 mmol of starting material. NMR spectra must be provided to assess purity. “after removing solvent” is not useful.

P15 : “1-{4-hydroxy-3-[(piperidin-1-yl) methyl] phenyl} ethan-1-one (4a)” : “mixture of hexane” did the authors used n-hexane or a mixture of isomer? Please clarify. 2.15 gm vs 2.15 g

To conclude, the microwave synthesis of mannich bases in only 30 minutes is an interesting work but it is presently poorly presented. The manuscript can not be published in Molecules in its present form. I recommend a strong rewriting of the paper with comparative tables for optimization conditions. NMR and MS spectra must be provided in supplementary data.

Author Response

Reviewer 1 (Please note that line numbers may change from one computer to another)

Comments and Suggestions for Authors

Comment 1: The paper by Ali et al deals with the microwave synthesis of mannich bases followed by single crystals X ray analysis of two compounds.

First of all, I would like to says that using microwave irradiation is not always green! In this particular case, the authors reduced the reaction time from 22h to 30 minutes (which is a great improvement) but they used different stoichiometry (1.5 eq for amine and formaldehyde vs only 1) compared to cited previous work (ref 38). Excess reagent is definitively not green. The title should be modified.

Response 1: We are grateful for these insightful comments. The title was reviewed in the light of the reviewer's comment and changed satisfactorily from "Regioselectivity of Mono- and Disubstituted 4-Hydroxyacetophenone Derivatives via Microwave-Assisted Mannich Reaction: Green synthesis, XRD and HS-analysis" to "Microwave-Assisted Synthesis of Mono- and Disubstituted 4-Hydroxyacetophenone Derivatives via Mannich Reaction: synthesis, XRD and HS-analysis". Please see lines 1-5 in the MS. More related changes done succeeding changing the title; "Abstract", please see lines 17-23 in the MS.  

Comment 2: In the title, authors speak about regioselectivity which is known for this kind of compound and classical. Moreover, regioselectivity is not discussed in the manuscript. The title should be modified.

Response 2: Authors agree. The title was changed as shown above in response 1. Please see also lines 1-5 in the MS.

Comment 3: At the end of the introduction, authors argue that the paper describe large scale reaction. In the synthesis section (P13), they present the synthesis of 2a which yield to 3g. In the paper by Wu et al (ref 38) they obtained 19g of the same compound. This is not large scale

Response 3: Authors agree on this comment. It was corrected according to the reviewer’s valuable comment. "Large scale" was changed to become "gram scale". Please see line 97 in the MS.  

Comment 4: P3 : general synthesis : authors says they run reactions in small scales which is highly imprecise (like large scale). Authors should precise the scale in grams (mg?) or mole. Moreover, authors have to mention that reactions were performed in sealed vessel to hemp non-specialist reader. were the small scales experiment performed with all the amine or only with morpholine. What is the fixed time used for optimization? Those points need to be clarified. Moreover, when dealing with optimization a table with all experiments is greatly appreciated.

Response 4: The authors completely agree with reviewer's comments. Thank you. We have changed the text following reviewer's suggestions. Please see "General synthesis" in the MS. Also, a table of all experiments with optimization is created. Please see table 1, lines 176 in the MS.

Comment 5: P3 : When using water authors obtained an “insoluble product”. Insoluble is not precise. Insoluble in water? In DCM? In DMSO? Authors should add “insoluble in classical organic solvents”.

Response 5: Authors agree. It was changed according to the reviewer suggestion from "insoluble product" to become "insoluble product in classical organic solvents." Lines 109 in the MS.

Comment 6: Excellent yield was obtained with dioxane, which is not a green solvent… (see above).

Response 6: We understand and appreciate the reviewer's opinion. However, it saves energy and time and therefore we still believe it is environmentally benign.

Comment 7: Page 4 : authors used 1H and 13C NMR to characterize compound. Spectra of all compounds must be provided in supplementary data to evaluate purity and to follow authors explanations on disappearance of signals.

Response 7: Thank you again for this observation. We do agree that supplementary data is important and therefore it was provided following the reviewer's suggestion.

Comment 8: P5 : “Comparing the conventional method to the microwave radiation, it was found that the reaction time decreased considerably from 22 h using thermal heating to 30 min in order to furnish  mono and or disubstituted Mannich bases 2a and 2b in overall yield 99.9%” : I don’t really understand the sentence. Is the overall yield for mono or disubstituted reaction? (“mono and or disubstituted” is not clear). How is calculated 99.9 %? A table with conditions and yield for “compound a” and “compound b” could help to understand “15 minutes overall yield 64, 72, 77%”.

Response 8: Thank you for this suggestion. Table 1 was created following reviewer's suggestion. The yield is calculate after the column chromatography work has been completed. It is overall yield. A table is inserted. Please see line 176 in the MS.

Comment 9 P5 : “In order to improve the products ratio, different reactant ratios were used in 1,4-dioxane at 120°C 98 (which was found to be the optimal reaction temperature) and 300 W.” When dealing with optimization a table is needed with time, temperature, solvent, stoichiometry, yield etc.

Response 9: Authors agree. A table of all experiments with optimization is created. Please see line 176 in the MS.

 Comment 10: P5 : “For the novel compounds 3b and 4b the yield under the same 104 conditions was 85.7% and 94.3%, respectively. It should be noticed that using diethylamine did not 105 undergo disubstituted Mannich reaction under the same conditions but only mono substituted 5a 106 was obtained in an excellent yield 99.7%” : Table is needed.

Response 10: The authors understand the value of this comment. Hence, table 1 was inserted and the phrase "in an excellent yield, 99.7%” was changed to become "in nearly quantitative yield". Please see lines 176 for table 1.

Comment 11: Diethylamine do not provide disubstituted compound. Authors should provide a tentative of explanation.

Response 11: We added an explanation as highlighted in P5. Please see lines 173-175.

Comment 12: P12 : materials and methods : mass spectrometer is not described (model).

Response 12: Authors agree. The mass spectrometer model was described. Please see lines 316-307 in the MS.

Comment 13: Authors describes acronyms (TLC, CC, VLC) and didn’t use them. Thinlyaer in two words.

Response 13: Thank you again for this observation. (CC, VLC) were deleted. However, TLC was kept because it was used in the MS (lines 109, 112, 163, 306 and 330).Thin layer was amended in the MS according to the reviewer comment. Please see line 305.

Comment 14: P13 : “without KBr” is not useful since authors used ATR technic.

Response 14: Authors agree. “without KBr” was deleted following the reviewer’s suggestion.

Comment 15: P13 : general synthesis of mannich bases : authors should precise sealed vessel instead of capped with rubber cap since high pressure. At first sight all the reactions were run on 16 mmol scale (ie 2.18 g). quantity of reactant should be included in experimental section and not only equivalents. Yields should be uniformized in all the manuscript. Sometime there is no digits after the decimal point, sometime one, sometime 2. Not sure that 2 digits is representative but authors have to choose between zero and one digit. Same remark with mass of product.

Response 15: Authors agree. "capped with a rubber cap" was replaced by "sealed vessel". Please see line 107 in the MS.  Other comments were taken in consideration in all the MS. Changes were highlighted in yellow color.

Comment 16: P13 : 1-{4-hydroxy-3-[(morpholin-4-yl) methyl] phenyl} ethan-1-one (2a) : 1H NMR description is different from ref 38. Spectra must be provided.

Response 16: Comment noticed. Spectra were provided in the supplementary data section.

Comment 17: P14 : “1-{4-hydroxy-3,5-bis[(morpholin-4-yl) methyl] phenyl} ethan-1-one (2b)” : 2b instead of 2a

Response 17: Authors agree. 2b was replaced by 2a.

Comment 18: There is a problem with the yield. 5.39 g correspond to 16.1 mmol (MW of 2b = 334.4) which is superior to the 16 mmol of starting material. NMR spectra must be provided to assess purity. “after removing solvent” is not useful.

Response 18: Authors agree.  It is a typing error. The correct yield is 5.32 g, please see line 347.  NMR spectra were provided in supplementary data section. “after removing solvent” was removed.

Comment 19: P15 : “1-{4-hydroxy-3-[(piperidin-1-yl) methyl] phenyl} ethan-1-one (4a)” : “mixture of hexane” did the authors used n-hexane or a mixture of isomer? Please clarify. 2.15 gm vs 2.15 g

Response 19: Authors agree. “mixture of hexane” was replaced by “hexanes” throughout the MS.  

Comment 20: To conclude, the microwave synthesis of mannich bases in only 30 minutes is an interesting work but it is presently poorly presented. The manuscript can not be published in Molecules in its present form. I recommend a strong rewriting of the paper with comparative tables for optimization conditions. NMR and MS spectra must be provided in supplementary data.

Response 20: All the valuable suggestions and comments of reviewer-1 have been taken in consideration. Changes, mainly, have been highlighted in yellow in the MS. For examples, the title was changed, and thereafter, the abstract has been amended to emphasize the purpose of using Microwave methodology. The results also reshaped to clarify the optimization. A table for optimization conditions has been created. The supplementary data is included as well.

Reviewer 2 Report

The manuscript looks very interesting and I consider it would be accepted for the journal if it could be revised carefully. It it suggested to be revised as follows: 1) the expression of introduction and experimental procedure were not clear, 2) the explanation on green synthesis looks unsatisfactory, 3) discussion how does the microwave pay assist on the reaction?

Author Response

Reviewer 2

Comments and Suggestions for Authors

The manuscript looks very interesting and I consider it would be accepted for the journal if it could be revised carefully. It it suggested to be revised as follows: 1) the expression of introduction and experimental procedure were not clear, 2) the explanation on green synthesis looks unsatisfactory, 3) discussion how does the microwave pay assist on the reaction?

Response: Authors are grateful for these valuable comments. The authors agree with the reviewer’s comments. The manuscript has been extensively revised and reshaped according to the reviewer's comments. All comments have been taken into account and corrected in the manuscript. Changes have been highlighted in yellow for convenience. For examples, the title was modified, the abstract has been amended to emphasize the purpose of using Microwave methodology. The introduction was changed in two places. The results also re-shaped to clarify the optimization. A table for optimization conditions has been created. The supplementary data (NMR and MS) were included as well.  

Reviewer 3 Report

The paper by Ali et al. deals Microwave-assisted Mannich reaction with 4-hydroxyacetophenone to yield regioselective mono- and/or disubstituted Mannich bases in the absence of any catalysts. Full characterization by modern methods (NMR, IR and EIMS) and structural determination by X-ray crystallographic analysis as well as HS-analysis for the obtained products 2a and 3a are explored. The one of the most important finding in this paper is the first synthetic approach of mono- and disubstituted Mannich bases of 4-hydroxyacetophenone with secondary amines. This finding should attract attention for the green organic synthesis. The reviewer thus agrees with the publication in the Molecules after considering the minor comment.

1. All product yields in the main text and “Materials and Methods” should be recorded without significant figures. In the case of 99.9% or over 99% yield, please record as “quantitative yield”.

2. As the readers are easy to understand the results, the starting material ratios and product yields for 2a-5a and 2b-4b should be provided in Scheme 2

3. Abstract, lines 7; some underlines should be deleted.

4. Please insert one space between a number and unit e.g. as “300 W” in Scheme 2.

5. Ref. 42; “English” should be corrected to “Engl.”.

Author Response

Reviewer 3

Comments and Suggestions for Authors

The paper by Ali et al. deals Microwave-assisted Mannich reaction with 4-hydroxyacetophenone to yield regioselective mono- and/or disubstituted Mannich bases in the absence of any catalysts. Full characterization by modern methods (NMR, IR and EIMS) and structural determination by X-ray crystallographic analysis as well as HS-analysis for the obtained products 2a and 3a are explored. The one of the most important finding in this paper is the first synthetic approach of mono- and disubstituted Mannich bases of 4-hydroxyacetophenone with secondary amines. This finding should attract attention for the green organic synthesis. The reviewer thus agrees with the publication in the Molecules after considering the minor comment.

Comment 1: All product yields in the main text and “Materials and Methods” should be recorded without significant figures. In the case of 99.9% or over 99% yield, please record as “quantitative yield”.

Response 1:  Authors are grateful for these valuable comments.  The manuscript has been revised according to reviewer's comments. Product yields was changed to become “quantitative yield” throughout the MS.

Comment 2:  As the readers are easy to understand the results, the starting material ratios and product yields for 2a-5a and 2b-4b should be provided in Scheme 2

Response 2:  Authors agree. New Table 1 was created to make it easy for the readers. Please see line 176 in the MS.   

 Comment 3:  Abstract, lines 7; some underlines should be deleted.

 Response 3: Authors agree. Noticed and amended.   

Comment 4:  4 Please insert one space between a number and unit e.g. as “300 W” in Scheme 2.

Response 4: Noticed and amended   

Comment 5:  Ref. 42; “English” should be corrected to “Engl.”.

Response 5: Noticed and amended. Thank you for your valuable time and effort to improve the paper. 

Round 2

Reviewer 1 Report

Authors have taken in consideration all the reviewers remarks to improve the manuscript.

NMR and MS spectra were provided in supplementary file, table was added etc.

Some minors’ remarks:

-        Add the ionization and detector type for the bruker mass spectrometer (probably a quadrupole but electron impact or ESI as ionization source?)

-        In my opinion, it is a good point to add NMR conditions (solvent, NMR frequency and temperature) in the title of NMR spectra in SI. Moreover, to help the reader, it could be good to include molecule drawing on the spectra in addition to its number.

Finally, I support publication in Molecules.

Author Response

Reviewer 1

Comments and Suggestions for Authors

Comment 1: Some minors’ remarks: Add the ionization and detector type for the bruker mass spectrometer (probably a quadrupole but electron impact or ESI as ionization source?)

 Response 1: Thank you for this suggestion. Following the reviewer's suggestion, "EI" was inserted in the sentence to become "Mass spectra were recorded on a Bruker mass spectrometer using EI, housed at the National University of Singapore."  Please see line 316.

Comment 2: In my opinion, it is a good point to add NMR conditions (solvent, NMR frequency and temperature) in the title of NMR spectra in SI. Moreover, to help the reader, it could be good to include molecule drawing on the spectra in addition to its number.

Response 2: Authors agree that it is a good addition. NMR solvent, molecule drawing, and their numbers have been added. More information is available on the right-hand side of each spectrum.  

Comment 3: Finally, I support publication in Molecules

Response 3:  Thank you